# Psychostimulants and opioids differentially influence the epigenetic modification of histone acetyltransferase and histone deacetylase in astrocytes

**Mayur Doke[1], Gurudutt Pendyala[2], Thangavel Samikkannu[1] ***

**1** Department of Pharmaceutical Sciences, Irma Lerma Rangel College of Pharmacy, Texas A&M University, Kingsville, Texas, United States of America, **2** Department of Anesthesiology, University of Nebraska Medical Center, Omaha, NE, United States of America

* thangavel@tamu.edu

**Data Availability Statement:** All relevant data are within the manuscript and S1, S2 Raw images and S1 Dataset.

## Abstract

Illicit drugs are known to affect central nervous system (CNS). Majorly psychostimulants such as cocaine, methamphetamine (METH) and opioids such as morphine are known to induce epigenetic changes of histone modifications and chromatin remodeling which are mediated by histone acetyltransferase (HAT) and histone deacetylase (HDAC). Aberrant changes in histone acetylation-deacetylation process further exacerbate dysregulation of gene expression and protein modification which has been linked with neuronal impairments including memory formation and synaptic plasticity. In CNS, astrocytes play a pivotal role in cellular homeostasis. However, the impact of psychostimulants and opioid mediated epigenetic changes of HAT/HADCs in astrocytes has not yet been fully elucidated. Therefore, we have investigated the effects of the psychostimulants and opioid on the acetylation-regulating enzymes- HAT and HDACs role in astrocytes. In this study, Class I and II HDACs and HATs gene expression, protein changes and global level changes of acetylation of H3 histones at specific lysines were analyzed. In addition, we have explored the neuroprotective "nootropic" drug piracetam were exposed with or without psychostimulants and opioid in the human primary astrocytes. Results revealed that psychostimulants and opioid upregulated HDAC1, HDAC4 and p300 expression, while HDAC5 and GCN5 expression were downregulated. These effects were reversed by piracetam coexposure. Psychostimulants and opioid exposure upregulated global acetylation levels of all H3Ks, except H3K14. These results suggest that psychostimulants and opioids differentially influence HATs and HDACs.

## 1. Introduction

Histone modification is one of the epigenetic alterations caused by extrinsic environmental factors without changing underlying DNA sequence that regulates gene activity [1]. These changes mainly induce chromatin remodeling, and subsequently causing alterations in genetic expression [2, 3]. Histone proteins in chromatin undergo post translational modification like

**Funding:** The present study was supported by grants from National Institute of Health (NIH)/ National Institute of Drug Abuse (NIDA): R01 DA044872 to S. Thangavel.

**Competing interests:** The authors declare that they have no conflicts of interest.

histone acetylation, phosphorylation, and methylation. This determines whether the transcriptional activity will be turned on or turned off, thus further deciding the fate of transcription of the downstream genes [2, 4]. Studies have also suggested that histone modifications, including HAT/HADCs, are critical regulators of gene expression by psychostimulants induction and known to affect several cellular functions including central nervous system (CNS) [5].

The psychostimulants and opioid mainly affect the CNS by disrupting the brain's cellular function, therefore affecting decision making and can cause memory dysfunction [6]. Additionally, psychostimulants and opioid can induce oxidative stress, energy metabolism and epigenetic modifications [7]. Furthermore, these drugs disrupt complex interactions in the brain linked to astrocytes that engage in crosstalk with neurons. These subsequently lead to a variety of CNS dysfunctions, including effects on neurogenesis and synaptic diversity [8]. Astrocytes play an essential role in energy metabolism, neurotransmission, and the inflammatory process [9]. Epigenetic modifications in the transcriptional machinery of astrocytes may drive astrocyte reactivity, thereby contributing to neuronal injury due to psychostimulant and opioid abuse [8]. Therefore, astrocytes are being recognized as promising targets for neuroprotective drugs [10].

Psychostimulant and opioid abuse has been shown to epigenetically modify histone proteins and to affect neuroplasticity *in vitro* and in animal models [11]. Histone acetyltransferases (HATs) acetylate histones by transferring acetyl groups from acetyl-CoA to lysine residues, while HDACs deacetylate lysine residues of histone proteins [12]. HATs consist of p300/CBP-associated factor (PCAF) and GCN5 belongs to Gcn5-related N-acetyltransferase (GNAT) family, whereas p300 belongs to CREB-binding protein (p300/CBP) coactivator family. Histone deacetylase (HDAC) class I and II proteins share a conserved catalytic domain and depend on $Zn^{2+}$ for catalytic activity [13]. Previous studies have shown that cocaine exposure causes imbalances in the levels of HDACs and HATs. This result into gene dysregulation due to the hypo- or hyperacetylation of genes [14–16]. Furthermore, histone acetylation is elevated by decreasing HDAC5 activity in the Nucleus Accumbens (NAc) and increasing the expression of HDAC5 target genes [17]. *In vivo* study has also shown that cocaine administration causes alteration in H4K12Ac and H3K9Ac levels and these changes was accompanied by an increase in HAT activity in the hippocampus region of the brain [18]. Preclinical studies have shown that the repeated self-administration of heroin results in an increase in global H3 acetylation within the mesolimbic dopamine system [19, 20]. Researchers have shown that repeated exposure to opioids causes hyperacetylation at H3K9, H3K14, H3K18, and H3K27 [21–25]. Studies have also revealed that methamphetamine (METH) exposure changes the levels of class I and II HDACs [26–28]. *In vivo* studies have displayed that METH acts as a dynamic epigenetic modifier that causes increase in H3 acetylation, hence, as a result it further modulates conditioned place preference (CPP) in animal model [29].

Structurally diverse HDAC inhibitors including pan-HDAC inhibitors and class-selective inhibitors have been evaluated for the treatment of neurodegenerative diseases [30, 31]. In addition, Kennedy *et al*. (2013) found that cocaine-induced plasticity changes can be blocked by the inhibition of class I HDACs [32]. Clinical studies have revealed that the "nootropic" drug piracetam (2-oxo-1-pyrrolidine acetamide), a cyclic derivative of the neurotransmitter γ-aminobutyric acid (GABA), can modulate cognitive function without causing sedation or stimulation [33, 34]. Piracetam is used as a treatment for Alzheimer's disease, dementia, memory dysfunction, alcoholism and brain injury [35].

To understand the effect of psychostimulant and opioid abuse on the enzymes governing histone acetylation, we performed mechanistic studies to assess the effects of cocaine, METH and morphine on class I and class II HDACs as well as HATs in astrocytes of the neuronal system. In this study, we investigated whether exposure to cocaine, METH and morphine has an

impact on HDACs 1–7 and HATs (PCAF, p300, GCN5) levels in astrocytes, which may further affect the acetylation levels of H3 histone proteins at various positions of lysine residues (H3K9AC, H3K14Ac, H3K18AC, H3K27AC, H3K56AC). We also studied the neuroprotective role of piracetam on HDAC 1–7 and HAT gene and protein expression in astrocytes.

## 2. Materials and methods

### 2.1 Cell culture and reagents

Psychostimulants–cocaine and METH, Opioid- morphine and piracetam (purity > 99%) were purchased from Sigma-Aldrich (CAS- St. Louis, MO, USA). Cell culture reagents were purchased from ScienCell (Carlsbad, CA, USA). Primary antibodies against HDACs (1–7) were purchased from Cell Signaling Technology, Inc. (Danvers, MA, USA). The HATs PCAF and GCN5 were purchased from Proteintech (Rosemont, IL) and p300 was purchased from Epigentek (Farmingdale, NY). Histone H3 Acetylation Antibody Panel Pack I (H3K9ac, H3K14ac, H3K18ac, H3K27ac) and H3K56ac were purchased from Epigentek (Farmingdale, NY, USA). A polyclonal histone-H3 antibody was purchased from Proteintech (Rosemont, IL). Electrophoresis reagents and nitrocellulose membranes were purchased from Bio-Rad (Richmond, CA, USA). All other reagents were purchased from Sigma-Aldrich (St. Louis, MO, USA).

### 2.2 Primary human astrocytes

In this study, human primary astrocytes (isolated from the cerebral cortex) were obtained from ScienCell (CAT-1800). Cultured cells were maintained in basal astrocyte medium supplemented with astrocyte growth factors (AGFs) and fetal bovine serum (FBS) at final concentrations of 10% and 1%, respectively, in antibiotic/antimycotic solution from ScienCell (Carlsbad, CA, USA). The human primary astrocytes for all treatment groups were counted using a microscope (100x) and viability always exceeded 95%, as determined by Trypan Blue exclusion method (Sigma-Aldrich—St. Louis, MO, USA).

### 2.3 Drug treatment

Piracetam, cocaine, METH and morphine were prepared in cell culture-grade distilled water to obtain working concentrations. To investigate the effects of cocaine, METH, morphine and piracetam, the cultured cells were divided into eight groups: (i) control cells exposed to media alone, (ii) piracetam (10 μM), (iii) cocaine (1 μM), (iv) piracetam and cocaine, (v) METH (10 μM), (vi) piracetam and METH, (vii) morphine (5 μM), and (viii) piracetam and morphine for 24 h. The doses used in this study were based from our previous published studies [36–39]. Cocaine (1 μM), METH (10 μM), morphine (5 μM) and piracetam (10 μM) did not show any toxicity at particular mentioned concentrations.

### 2.4 RNA extraction and quantitative real time polymerase chain reaction (PCR) messenger RNA (mRNA) assay

Total RNA was extracted from the different treatment groups using Qiagen RNeasy Mini Kit catalog no. 74104 (Germantown, MD, USA). Absorbance at 260 nm was measured to evaluate the purity of RNA by NanoDrop spectrophotometer. Total RNA was screened for purity with 260/280 ratio of ~2.0 and cDNA was synthesized from pre-screened total RNA using iScript™ cDNA Synthesis Kit (Bio-Rad Richmond, CA, USA). The amplification of cDNA was performed using specific primers for HDAC1 (Hs02621185_s1), HDAC3 (Hs00187320_s1), HDAC4 (Hs01041648_m1), HDAC6 (Hs00997427_m1), HDAC7 (Hs05599594_g1), GCN5/

KAT2A (Hs00221499_m1), PCAF/KAT2B (Hs00187332_m1) and β-actin (Hs99999903_m1) (Applied Biosystems, Foster City, CA). For quantification of real time PCR, β-actin was used as internal control. Initial denaturation was done at 95˚C for 2 min followed by denaturation (40 cycles) at 94˚C for 15 s. Annealing was done at 59˚C for 1 min and extension follows by at 72˚C for 15 s. Relative mRNA expression was quantified and $2^{-\Delta\Delta CT}$ was used to calculate fold change in expression of target gene. Results of RNA from treated samples were normalized to results from control (untreated) samples. To ensure reproducibility of data, all experiments were performed from a minimum of three biological replicates.

## 2.5 Western blot analysis

A total protein lysate containing HDACs (1–7) was purchased from Cell Signaling Technology, Inc., the HATs PCAF and GCN5 were purchased from Proteintech, and p300 was purchased from Epigentek for expression analysis by western blotting. Cells were collected and washed twice in 1X PBS and then lysed in ice-cold lysis buffer with M-PER™ (Mammalian Protein Extraction Reagent, Thermo Scientific, Waltham, MA, USA) on ice for 1 h. The cell lysates were then centrifuged for 15 min at 13,000 rpm at 4˚C. An equal amount of protein was resolved by 4–15% gradient polyacrylamide gel electrophoresis (SDS-PAGE) and subsequently transferred to a nitrocellulose membrane. Antibody dilutions used were according to manufacturer's recommendations for detection by immunoblot. Membranes were then incubated with a peroxidase-conjugated secondary antibody for 1 h. Immunoreactive bands were visualized ECL Plus Western blot reagents (Bio-Rad laboratories, USA). Electrochemiluminescence (ECL) intensity of detected target proteins was imaged and quantified with a c300 (azure biosystems) and densitometric analysis was carried out using ImageJ digitalizing software.

## 2.6 Histone extraction

Human primary astrocytes exposed to the psychostimulants—cocaine, METH and Opioid-morphine were harvested, pelleted, and washed twice with ice-cold PBS. Histone proteins were isolated from cells using EpiQuik total histone extraction kit and further process followed as per the manufacturer's protocol (EpiGentek, Farmingdale, NY). The protein concentration was measured via the Bradford Protein Assay, and the protein was stored at -20˚C until further use.

## 2.7 Immunofluorescence staining

For immunofluorescence studies of HDAC1 and GCN5, cells were treated with the psychostimulants and opioid with or without piracetam and grown in chamber slides. For immunostaining cells were fixed with 4% paraformaldehyde, followed by permeabilization with 0.2% Triton X-100 in PBS for 15 min at room temperature and then blocked with 5% normal goat serum at 4˚C for 1h and then incubated with antibodies: anti-HDAC1 (Cell Signaling Technology) and anti-GCN5 (Proteintech) overnight at 4˚C. After washing with 1X PBS, cells were incubated with anti-mouse-IgG Alexa Fluor® 633 for GCN5 and anti-rabbit-IgG Alexa Fluor® 546 for HDAC1 for 2h. The cellular nuclei were stained with 4'6-diamidino-2-phenylindole (DAPI). Slides were examined with a Nikon C1plus confocal microscope.

## 2.8 Statistical analysis

Statistical analysis was performed using GraphPad Prism version 6. Differences between the control and treatment groups- piracetam-only, cocaine-only, cocaine + piracetam, METH only, METH + Piracetam, Morphine only and Morphine + Piracetam were calculated using

two-way ANOVA followed by Tukey's post hoc test. We have also compared cocaine, METH and morphine with cocaine + piracetam, METH + Piracetam, and Morphine + Piracetam respectively. Values are expressed as the mean ± standard error mean of three independent experiments, and a significance level of $p < 0.05$ was used for western blot protein analyses. In case of RT-PCR gene expression analysis, differences between control and treatment groups were calculated using two-way ANOVA followed by Tukey's post hoc test and expressed as the mean ± standard error mean of three independent experiments, and a significance level of $p < 0.05$ was used. To analyze the immunofluorescence data, corrected total cell fluorescence (CTCF) values were determined using Fiji-ImageJ [40], and statistical comparisons were performed using one-way ANOVA followed by Dunnett's post hoc test. Values are expressed as the mean ± standard error mean of three independent experiments, and a significance level of $p < 0.05$ was used.

The following formula was used to calculate CTCF:

$$CTCF = \text{Integrated Density} \\ - (\text{area of selected cell X mean fluorescence of background readings})$$

## 3. Results

### 3.1 Effect of cocaine, METH, morphine and the nootropic drug piracetam on class I HDAC protein levels in human primary astrocytes

HDAC class I proteins are ubiquitously expressed in brain tissue and play a major role in the deacetylation of histone proteins. To examine the effects of cocaine, METH and morphine on the expression of HDACs (1–3), human primary astrocytes were exposed to cocaine (1 μM), METH (10 μM), and morphine (5 μM) for 24 h. The exposure of astrocytes to cocaine ($F_{(7, 14)}$ = 15.16, P = 0.0105) and METH ($F_{(7, 14)}$ = 15.16, P = 0.0150) significantly increased HDAC1 protein levels, while there was no significant increase in HDAC1 protein levels when astrocytes were exposed to morphine compared to those in the control (Fig 1A and 1D). Additionally, we investigated the effect of the nootropic drug piracetam on class I HDAC protein levels. Interestingly, coexposure to the psychostimulants and piracetam restored the HDAC1 proteins to levels similar to those observed in the control (Fig 1A and 1D). We furher analyzed to compare the groups between cocaine Vs cocaine + piracetam ($F_{(7, 14)}$ = 15.16, P = 0.0004), METH Vs METH + piracetam ($F_{(7, 14)}$ = 15.16, P <0.0001) and morphine Vs morphine + piracetam (NS) for HDAC1. Furthermore, we observed no significant changes in HDAC2 protein levels when astrocytes were exposed to cocaine and METH with or without coexposure with piracetam (Fig 1B and 1E). However, we observed a significant decrease in the HDAC2 protein level ($F_{(7, 14)}$ = 17.53, P = 0.0017) under the exposure of astrocytes to morphine (Fig 1B and 1E). Interestingly, coexposure to morphine with piracetam increased the HDAC2 protein level compared to that observed under morphine exposure alone and prevented HDAC2 protein inhibition (Fig 1B and 1E). We analyzed to compare the groups between cocaine Vs cocaine + piracetam ($F_{(7, 14)}$ = 17.53, P = 0.0450), METH Vs METH + piracetam ($F_{(7, 14)}$ = 17.53, P = 0.0006) and morphine Vs morphine + piracetam ($F_{(7, 14)}$ = 17.53, P = 0.0010) for HDAC2. Furthermore, we investigated the epigenetic effect of cocaine, METH and morphine on HDAC3 protein levels. We did not observe any significant changes in HDAC3 protein levels compared to those in the control when astrocytes were exposed to cocaine, METH and morphine (Fig 1C and 1F). We then the groups between cocaine Vs cocaine + piracetam ($F_{(7, 14)}$ = 22.19, P = 0.037), METH Vs METH + piracetam ($F_{(7, 14)}$ = 22.19, <0.0001) and morphine Vs morphine + piracetam (NS) for HDAC3. Fig 1D–1F show the densitometric values representing HDAC1, HDAC2 and HDAC3 protein levels (fold-change compared to the control), respectively. In order to have a deeper understanding of the effect of the psychostimulants on

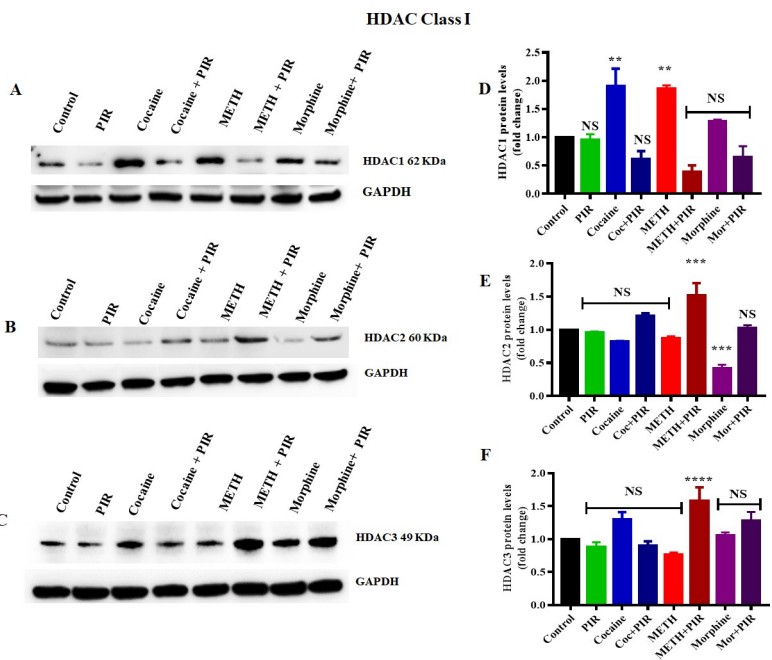

**Fig 1. Effects of the psychostimulants cocaine, METH and morphine on HDAC class I proteins in human primary astrocytes.** Cells were exposed to cocaine (1 μM), METH (10 μM), and morphine (5 μM) either alone or in combination with piracetam (10 μM) for 24 h. The protein expression levels of different classes of HDACs (1–3) in astrocytes were determined by western blotting analysis using GAPDH as a loading control. Western blot showing (A) HDAC1, (B) HDAC2 and (C) HDAC3. The densitometric analysis results in D, E and F represent the protein levels (fold-change control) of HDAC1, HDAC2 and HDAC3, respectively. Two-way ANOVA analysis performed to compare the groups between cocaine Vs cocaine + piracetam, METH Vs METH + piracetam and morphine Vs morphine + piracetam. The data are expressed as the mean ± standard error mean of three independent experiments. N = 3. ****P<0.0001, ***P<0.001, **P<0.01, NS—nonsignificant.

the protein levels of class II histone deacetylase enzymes, we evaluated the effects of cocaine, METH and morphine on class II HDAC proteins.

## 3.2 Effect of cocaine, METH, morphine and the nootropic drug piracetam on class II HDAC protein levels in human primary astrocytes

The HDAC class II proteins comprise HDACs 4, 5, 6, 7, 9, & 10 and are localized in both the nucleus and the cytoplasm. Studies have investigated the roles of Class II HDACs in the responses to numerous environmental stimuli and neural signaling pathways. However, the epigenetic changes in class II HDAC protein levels induced by cocaine, METH and morphine have not yet been studied under similar conditions in human primary astrocytes. We examined the effects of cocaine, METH and morphine on the protein levels of HDACs (4, 5, 6 and 7). Human primary astrocytes were exposed to cocaine (1 μM), METH (10 μM), and morphine (5 μM) for 24 h. The exposure of astrocytes to METH ($F_{(7, 14)}$ = 9.765, P = 0.0079) and morphine ($F_{(7, 14)}$ = 9.765, P = 0.0026) significantly increased HDAC4 protein levels compared to those in the control (Fig 2A and 2E). Exposure to cocaine also upregulated HDAC4 protein levels compared to those in the control, but the changes were not significant (Fig 2A and 2E). Remarkably, we observed that the coexposure of astrocytes to piracetam with morphine reversed the psychostimulant effect on HDAC4 levels, resulting in an expression level equivalent to that in the control group, as shown in Fig 2A and 2E. We analyzed to compare the groups between cocaine Vs cocaine + piracetam (NS), METH Vs METH + piracetam (NS)

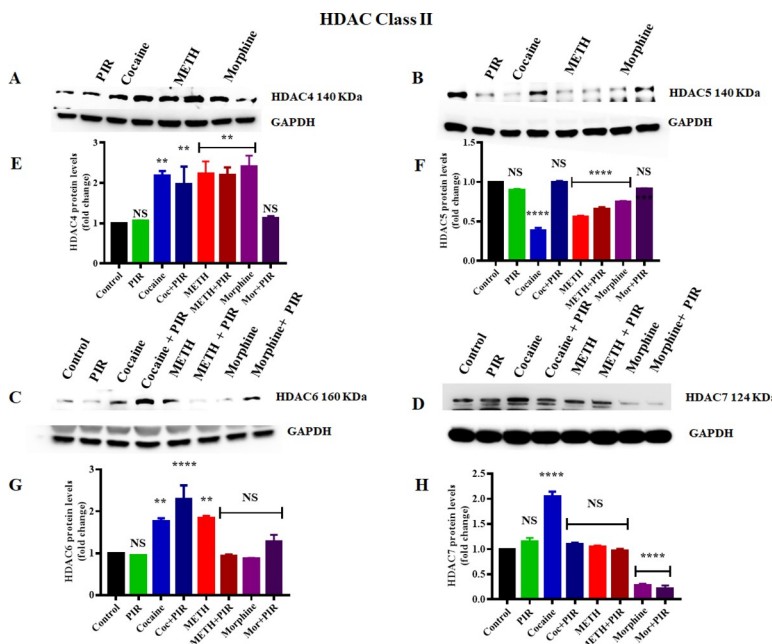

**Fig 2. Effects of the psychostimulants cocaine, METH and morphine on HDAC class II proteins in human primary astrocytes.** Cells were exposed to cocaine (1 μM), METH (10 μM), and morphine (5 μM) either alone or in combination with piracetam (10 μM) for 24 h. The protein expression levels of different classes of HDACs (4, 5, 6 and 7) in astrocytes were determined by western blotting analysis using GAPDH as a loading control. Western blot showing (A) HDAC4, (B) HDAC5, (C) HDAC6 and (D) HDAC7. Two-way ANOVA analysis performed to compare the groups between cocaine Vs cocaine + piracetam, METH Vs METH + piracetam and morphine Vs morphine + piracetam. The densitometric analysis results in E, F, G and H represent the protein levels (fold-change control) of HDAC4, HDAC5, HDAC6 and HDAC7, respectively. The data are expressed as the mean ± standard error mean of three independent experiments. N = 3. ****$P<0.0001$, ***$P<0.001$, **$P<0.01$, NS—nonsignificant.

and morphine Vs morphine + piracetam (F$_{(7, 14)}$ = 9.765, P = 0.063) for HDAC4. Similarly, we investigated the effects of the psychostimulants on HDAC5 protein levels. We observed that the exposure of human primary astrocytes to cocaine, METH and morphine significantly decreased the HDAC5 protein level (F$_{(7, 14)}$ = 155.4, P< 0.0001) compared to that in the control, as shown in Fig 2B and 2F. Intriguingly, we observed that the coexposure of astrocytes to piracetam and cocaine or morphine reversed the psychostimulant effect on HDAC5 protein levels and increased HDAC5 expression to levels equal to those in the control (Fig 2B and 2F). We analyzed to compare the groups between cocaine Vs cocaine + piracetam (F$_{(7, 14)}$ = 155.4, P<0.0001), METH Vs METH + piracetam (F$_{(7, 14)}$ = 155.4, P = 0.0326) and morphine Vs morphine + piracetam (F$_{(7, 14)}$ = 155.4, P = 0.0002) for HDAC5.

We also investigated the effect of cocaine, METH and morphine on the protein levels of HDAC6 and HDAC7. We observed that cocaine significantly upregulated HDAC6 protein levels (F$_{(7, 14)}$ = 16.25, P = 0.0183) and HDAC7 protein levels (F$_{(7, 14)}$ = 104.6, P<0.0001) compared to those in the control, as shown in Fig 2C and 2D, respectively. Morphine exposure did not have any significant effect on HDAC6 protein levels, while HDAC7 protein levels (F$_{(7, 14)}$ = 104.6, P<0.0001) were significantly downregulated compared to those in the control (Fig 2C and 2D). Coexposure to cocaine and piracetam decreased the HDAC7 protein level and reversed to the level of the control group, as shown in Fig 2D and 2H. However, coexposure to cocaine and piracetam increased the HDAC6 protein level (F$_{(7, 14)}$ = 16.25, P = 0.0001) compared to that in the control, as shown in Fig 2C and 2G. We analyzed to compare the groups between cocaine Vs cocaine + piracetam (NS), METH Vs METH + piracetam (F$_{(7, 14)}$ = 16.25,

P = 0.0046) and morphine Vs morphine + piracetam (NS) for HDAC6. We analyzed to compare the groups between cocaine Vs cocaine + piracetam ($F_{(7, 14)}$ = 104.6, P <0.0001), METH Vs METH + piracetam (NS) and morphine Vs morphine + piracetam (NS) for HDAC7. These results suggest that cocaine, METH and morphine epigenetically affect class II HDAC protein levels in strikingly different manners and that the protective effects of piracetam may help to counteract the impacts of different psychostimulants and opioid on deacetylation activity in human primary astrocytes. Fig 2E–2H show the densitometric values representing HDAC4, HDAC5, HDAC6 and HDAC7 protein levels (fold-change compared to the control), respectively.

## 3.3 Effect of cocaine, METH, morphine and the nootropic drug piracetam on HAT protein levels in human primary astrocytes

The balance between deacetylation and acetylation of histone proteins are maintained by HATs which counteract the effects of HDACs. Previous studies have investigated the psychostimulant and opioid-induced alteration in the histone acetyltransferase enzymes. Changes in HATs expression can mediate acetylation of early genes which can further alter neuroplasticity associated with psychostimulant and opioid addiction [41, 42]. However, there is no evidence of the role of astrocyte-specific HATs in histone acetylation responses to cocaine, METH and morphine. Therefore, we exposed human primary astrocytes to cocaine (1 μM), METH (10 μM), and morphine (5 μM) for 24 h and determined the effect of the psychostimulants on the levels of the HAT proteins GCN5, PCAF and P300, which acetylate histone H3 and H4 and are expressed in the brain [43].

We observed that cocaine exposure significantly downregulated PCAF ($F_{(7, 14)}$ = 52.45, P = 0.0001) and GCN5 ($F_{(7, 14)}$ = 15.99, P = 0.0018) protein levels, while p300 ($F_{(7, 14)}$ = 91.85, P = 0.0001) protein levels were upregulated (Fig 3A–3C). Moreover, we also observed that morphine exposure significantly downregulated PCAF ($F_{(7, 14)}$ = 52.45, P = 0.0001) protein levels compared to those in the control. Morphine exposure in human primary astrocytes slightly decreased GCN5 protein levels, but the difference compared to the control levels was not significant, as shown in Fig 3C and 3F. Interestingly, coexposure to morphine and piracetam restored the protein levels of GCN5 compared to those in the control, as shown in Fig 3C. Additionally, we investigated the effect of METH on HAT protein levels and observed that METH significantly upregulated p300 levels ($F_{(7, 14)}$ = 91.85, P = 0.0001); on the other hand, METH significantly downregulated GCN5 ($F_{(7, 14)}$ = 15.99, P = 0.0210) and PCAF ($F_{(7, 14)}$ = 52.45, P = 0.0001) levels compared to those in the control, as shown in Fig 3B–3E, 3C–3F and 3A–3D. We analyzed to compare the groups between cocaine Vs cocaine + piracetam (NS), METH Vs METH + piracetam (NS) and morphine Vs morphine + piracetam ($F_{(7, 14)}$ = 52.45, P <0.0001) for PCAF. We analyzed to compare the groups between cocaine Vs cocaine + piracetam (NS), METH Vs METH + piracetam (NS) and morphine Vs morphine + piracetam (NS) for p300. We analyzed to compare the groups between cocaine Vs cocaine + piracetam (NS), METH Vs METH + piracetam ($F_{(7, 14)}$ = 15.99, P = 0.0007) and morphine Vs morphine + piracetam ($F_{(7, 14)}$ = 15.99, P = 0.0023) for GCN5. Coexposure with piracetam and METH restored the GCN5 expression to levels to those in the control as shown in Fig 3C. Fig 3D–3F show the densitometric values indicating PCAF, p300 and GCN5 protein levels (fold-change compared to the control).

## 3.4 Effect of cocaine, METH, morphine and the nootropic drug piracetam on H3 acetylation protein levels in human primary astrocytes

Histone posttranscriptional modifications have shown to regulate downstream gene expression by inducing chromatin remodeling and alteration of transcription status [44, 45]. Among

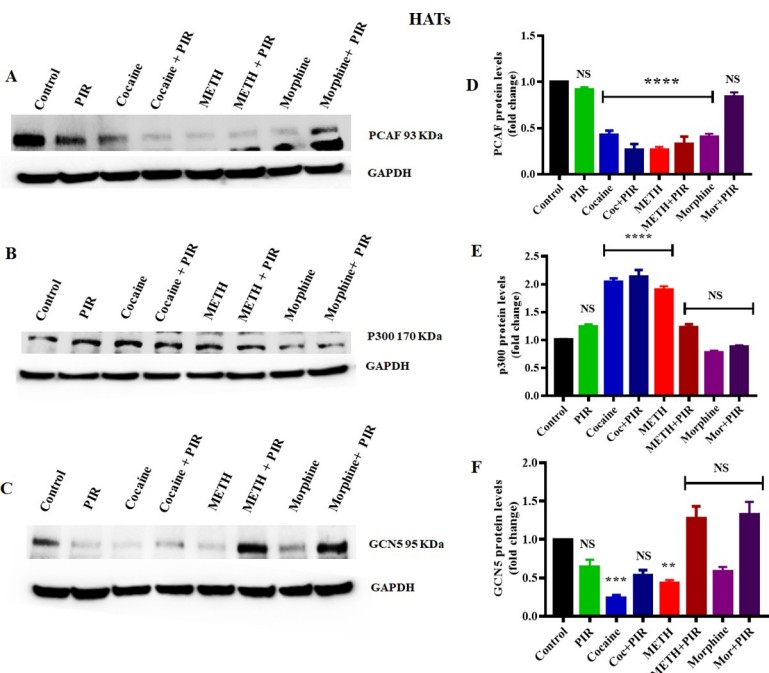

**Fig 3. Effects of the psychostimulants cocaine, METH and morphine on HAT proteins in human primary astrocytes.** Cells were exposed to cocaine (1 μM), METH (10 μM), and morphine (5 μM) either alone or in combination with piracetam (10 μM) for 24 h. The protein expression levels of PCAF, p300 and GCN5 in astrocytes were determined by western blotting analysis using GAPDH as a loading control. Two-way ANOVA analysis performed to compare the groups between cocaine Vs cocaine + piracetam, METH Vs METH + piracetam and morphine Vs morphine + piracetam. Western blot showing (A) PCAF, (B) p300 and (C) GCN5. The densitometric analysis results in D, E and F represent the protein levels (fold-change control) of PCAF, p300 and GCN5, respectively. The data are expressed as the mean ± standard error mean of three independent experiments. N = 3. ****P<0.0001, ***P<0.001, **P<0.01, NS—nonsignificant.

the five histones, H3 is considered to be the most extensively modified [46]. Therefore, we investigated the effects of cocaine, METH and morphine on the acetylation levels of histone H3 lysines 9, 14, 18, 27, and 56 (H3K9, H3K14, H3K18, H3K27, and H3K56). Our results revealed that cocaine and METH slightly increased the H3K9AC level, but the change was not significant. However, morphine significantly increased the H3K9AC level (F$_{(7, 14)}$ = 28.61, P = 0.0029) compared to that in the control, and coexposure to piracetam and morphine significantly decreased (F$_{(7, 14)}$ = 28.61, P = 0.0001) the H3K9AC level compared to that in the control (Fig 4A and 4G). We analyzed to compare the groups between cocaine Vs cocaine + piracetam (NS), METH Vs METH + piracetam (NS) and morphine Vs morphine + piracetam (F$_{(7, 14)}$ = 28.61, P <0.0001) for H3K9AC. In the case of the H3K14AC level, we observed that cocaine and METH exposure resulted in significant downregulation (F$_{(7, 14)}$ = 117.5, P = 0.0001), while morphine exposure resulted in significant upregulation (F$_{(7, 14)}$ = 117.5, P = 0.0001) compared to the level in the control (Fig 4B and 4H). Coexposure to cocaine or METH and piracetam did not result in any significant change compared to those observed under exposure to either cocaine or METH alone, respectively (Fig 4B and 4H). However, coexposure to morphine and piracetam in astrocytes reversed H3K14AC expression to a level similar to that in the control (Fig 4B and 4H). We analyzed to compare the groups between cocaine Vs cocaine + piracetam (NS), METH Vs METH + piracetam (F$_{(7, 14)}$ = 117.5, P = 0.0005) and morphine Vs morphine + piracetam (F$_{(7, 14)}$ = 117.5, P <0.0001) for H3K14AC. Furthermore, our results revealed that H3K18AC levels changed significantly upon

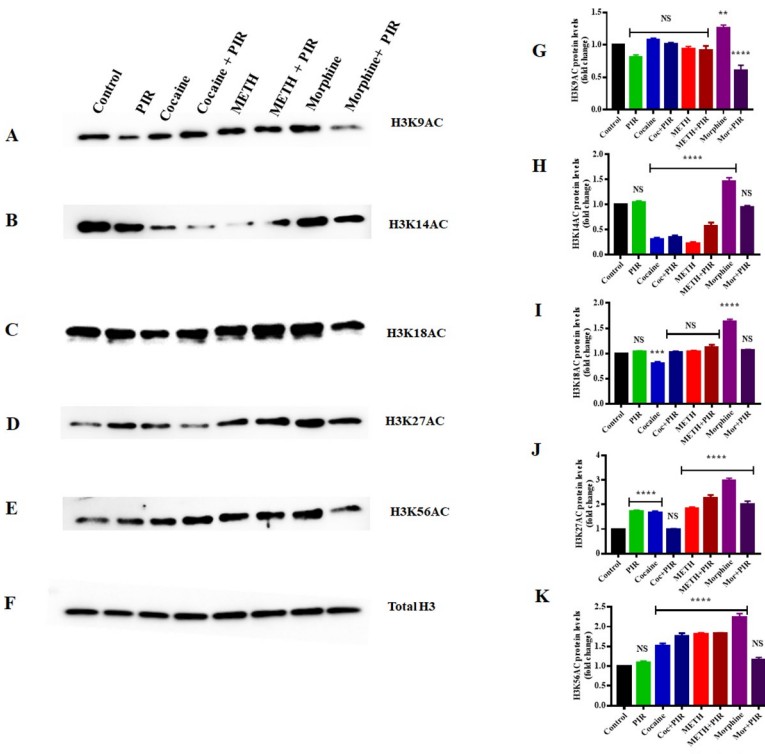

**Fig 4. Effects of the psychostimulants cocaine, METH and morphine on H3 histone proteins in human primary astrocytes.** Cells were exposed to cocaine (1 μM), METH (10 μM), and morphine (5 μM) either alone or in combination with piracetam (10 μM) for 24 h. The protein expression levels of H3K9AC, H3K14AC, H3K18AC, H3K27AC, and H3K56AC in astrocytes were determined by western blotting analysis using total H3 as a loading control. Two-way ANOVA analysis performed to compare the groups between cocaine Vs cocaine + piracetam, METH Vs METH + piracetam and morphine Vs morphine + piracetam. Western blot showing (A) H3K9AC, (B) H3K14AC, (C) H3K18AC, (D) H3K27AC, and (E) H3K56AC. The densitometric analysis results in G, H, I, J and K represent the protein levels (fold-change control) of H3K9AC, H3K14AC, H3K18AC, H3K27AC, and H3K56AC, respectively. The data are expressed as the mean ± standard error mean of three independent experiments. N = 3. ****$P<0.0001$, ***$P<0.001$, **$P<0.01$, NS—nonsignificant.

exposure to cocaine (F $_{(7, 14)}$ = 74.45, P = 0.0001) and morphine (F $_{(7, 14)}$ = 74.45, P = 0.0001), while METH exposure did not result in any change compared to the control levels (Fig 4C and 4I). Interestingly, the coexposure of astrocytes to cocaine and morphine with piracetam reversed the H3K18AC level to the control level (Fig 4C and 4I).

We analyzed to compare the groups between cocaine Vs cocaine + piracetam (F $_{(7, 14)}$ = 74.45, P = 0.0012), METH Vs METH + piracetam (NS) and morphine Vs morphine + piracetam (F $_{(7, 14)}$ = 74.45, P <0.0001) for H3K18AC. Moreover, we investigated the acetylation level of H3K27. We observed that cocaine, METH and morphine significantly upregulated (F $_{(7, 14)}$ = 70.62, P = 0.0001) H3K27AC levels compared to those in the control (Fig 4D and 4J), and the effect of cocaine exposure was reversed by coexposure to cocaine and piracetam (Fig 4D and 4J). We analyzed to compare the groups between cocaine Vs cocaine + piracetam (F $_{(7, 14)}$ = 70.62, P = 0.0003), METH Vs METH + piracetam (F $_{(7, 14)}$ = 70.62, P = 0.0334) and morphine Vs morphine + piracetam (F $_{(7, 14)}$ = 70.62, P <0.0001) for H3K27AC. In the case of H3K56AC, our data revealed that cocaine, METH and morphine significantly upregulated (F $_{(7, 14)}$ = 74.51, P = 0.0001) H3K56AC levels (Fig 4E and 4K) and that coexposure to morphine and piracetam reversed the H3K56AC level to the control level (Fig 4E and 4K). We analyzed to compare the groups between cocaine Vs cocaine + piracetam (NS), METH Vs METH

+ piracetam (NS) and morphine Vs morphine + piracetam ($F_{(7, 14)}$ = 74.51, P <0.0001) for H3K56AC. Fig 4F shows the pan (total) H3 used as the loading control. Fig 4G–4K show the densitometric values indicating H3K9AC, H3K14AC, H3K18AC, H3K27AC, and H3K56AC protein levels (fold-change compared to the control), respectively.

### 3.5 Effect of cocaine, METH, morphine and the nootropic drug piracetam on HDAC class I, HDAC class II and HAT gene expression in human primary astrocytes

We primarily evaluated the effect of cocaine, METH, morphine and the nootropic drug piracetam on HDAC class I, HDAC class II and HATs at the protein level. In a further study, we focused our research on the validation of the effect of psychostimulants on HDAC class I, HDAC class II and HAT gene expression at the mRNA level. To gain insights into the molecular mechanism by which piracetam affects the expression of HDACs, astrocytes were exposed to cocaine (1 μM), METH (10 μM) and morphine (5 μM) either alone or in combination with piracetam (10 μM) for 24 h. Fig 5A shows that exposure to cocaine ($F_{(7, 14)}$ = 16.21, P = 0.0042), METH ($F_{(7, 14)}$ = 16.21, P = 0.0041) and morphine ($F_{(7, 14)}$ = 16.21, P = 0.0002) upregulated the mRNA expression of HDAC1, whereas coexposure to these psychostimulants and piracetam restored HDAC1 mRNA expression levels in a drug-dependent manner (Fig 5A). We analyzed to compare the groups between cocaine Vs cocaine + piracetam ($F_{(7, 14)}$ = 16.21, P = 0.0047), METH Vs METH + piracetam ($F_{(7, 14)}$ = 16.21, P = 0.0031) and morphine Vs morphine + piracetam ($F_{(7, 14)}$ = 16.21, P = 0.0007) for HDAC1. Similarly, we investigated the effect of cocaine, METH and morphine on HDAC3 mRNA levels. Our data revealed that cocaine and METH exposure did not have any significant effect on HDAC3 mRNA expression, whereas morphine significantly ($F_{(7, 14)}$ = 15.57, P = 0.0093) upregulated HDAC3 mRNA expression compared to that in the control, as shown in Fig 5B. Moreover, we examined the effect of exposure to cocaine, METH and morphine on HDAC class II gene expression. We observed that HDAC4 mRNA expression was not significantly changed upon exposure to cocaine and METH while morphine exposure changed ($F_{(7, 14)}$ = 6.17, P = 0.042) compared to the level in the control (Fig 5C).

Furthermore, our data revealed that METH ($F_{(7, 14)}$ = 25.24, P<0.0001) and morphine ($F_{(7, 14)}$ = 25.24, P = 0.028) exposure in astrocytes significantly upregulated HDAC6 mRNA expression, whereas cocaine exposure did not induce a significant change in HDAC6 mRNA expression compared to the level in the control (Fig 5D). Exposure to cocaine, METH and morphine did not induce any significant change in HDAC7 mRNA expression levels compared to those in the control (Fig 5E). We analyzed to compare the groups between METH Vs METH + piracetam ($F_{(7, 14)}$ = 25.24, P <0.0001) for HDAC6.

We also investigated the effect of cocaine, METH and morphine on GCN5 mRNA levels. We observed that cocaine, METH and morphine did not result in any significant change in GCN5 mRNA levels (Fig 5F). We analyzed to compare the groups between cocaine Vs cocaine + piracetam ($F_{(7, 14)}$ = 8.383, P = 0.0176), METH Vs METH + piracetam ($F_{(7, 14)}$ = 8.383, P = 0.0486) and morphine Vs morphine + piracetam (NS) for GCN5. In the case of PCAF, exposure to METH ($F_{(7, 14)}$ = 9.912, P = 0.0238) and morphine ($F_{(7, 14)}$ = 9.912, P = 0.0377) downregulated PCAF gene expression. We analyzed to compare the groups between cocaine Vs cocaine + piracetam ($F_{(7, 14)}$ = 9.912, P = 0.0028) for PCAF. These results show that cocaine, METH and morphine induce or modulate HDAC and HAT gene expression at the mRNA level, which may be implicated in the alteration of the acetylome of astrocytes. This may further impact the transcriptional machinery of astrocytes, which regulates the expression of genes.

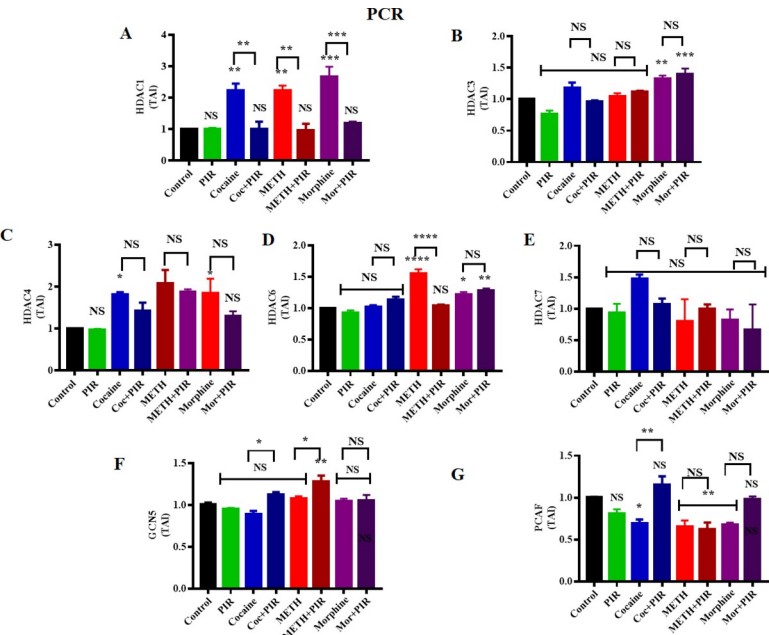

**Fig 5. Effects of the psychostimulants cocaine, METH and morphine on HDAC and HAT gene expression in human primary astrocytes.** Human primary astrocytes were exposed to cocaine (1 μM), METH (10 μM) and morphine (5 μM) either alone or in combination with piracetam (10 μM) for 24 h. Controls were maintained in drug-free medium (without drug exposure). (A) HDAC1, (B) HDAC3, (C) HDAC4, (D) HDAC6 and (E) HDAC7 (F) GCN5 (G) PCAF mRNA expression levels in astrocytes determined by qRT-PCR analysis using the housekeeping gene β-actin as a loading control. Two-way ANOVA analysis performed to compare the groups between cocaine Vs cocaine + piracetam, METH Vs METH + piracetam and morphine Vs morphine + piracetam. The data are expressed as the mean ± standard error mean of the transcript accumulation index (TAI) of three independent experiments. N = 3. ****P<0.0001, ***P<0.001, **P<0.01, *P<0.05, NS—nonsignificant.

## 3.6 The psychostimulants cocaine, METH, and morphine impact the nuclear localization of HDAC1, and piracetam exerts a protective effect

HDAC1 predominantly localizes to the nucleus. We found that HDAC1 was overexpressed upon exposure to cocaine, METH and morphine. However, combined exposure to piracetam protected cocaine-, METH- and morphine-induced HDAC1 overexpression (Fig 5A). Immunostaining analysis was performed to confirm the cocaine-, METH- and morphine-mediated overexpression and cellular localization of HDAC1 in astrocytes. The results obtained for the control group of astrocytes demonstrated that HDAC1 mainly exists in the nucleus. The translocation of HDAC1 from the nuclear to cytoplasmic compartments was substantially higher in cocaine- and METH-exposed cells than in the control. Moreover, HDAC1 expression and nuclear localization were restored by piracetam (Fig 6A and 6B). HDAC1 expression was significantly higher under cocaine ($F_{(7, 16)} = 10.58$, P = 0.0009), METH ($F_{(7, 16)} = 10.58$, P = 0.0002) and morphine ($F_{(7, 16)} = 10.58$, P = 0.0155) treatment than in the control (Fig 6A and 6B). We also examined GCN5 expression, which was mainly observed in the cytoplasm upon immunostaining analysis, and we observed that cocaine, METH and morphine downregulated the expression of GCN5 compared to that in the control, as shown in Fig 6C and 6D.

## 4. Discussion

The abuse of psychostimulants such as cocaine, METH and opioid such as morphine are known to induce the dysregulation of cellular functions, resulting in altered mitochondrial

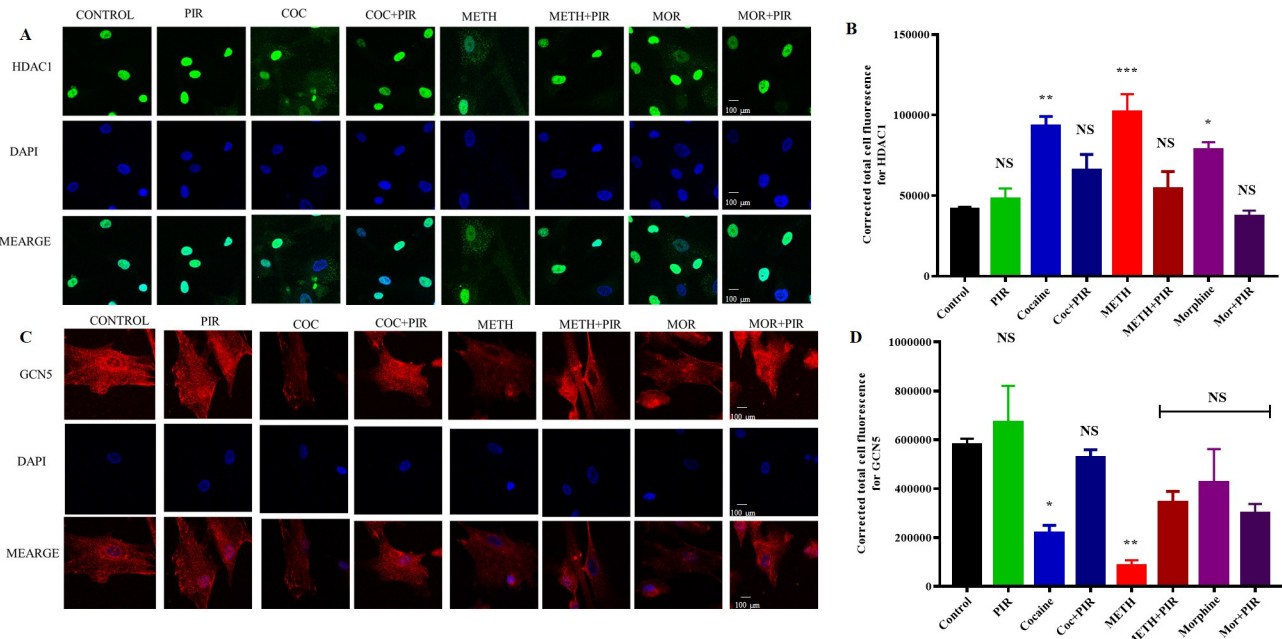

**Fig 6. Effects of the psychostimulants cocaine, METH and morphine on the expression and translocation of HDAC1 and GCN5 in human primary astrocytes.** Human primary astrocytes were exposed to cocaine (1 µM), METH (10 µM) and morphine (5 µM) either alone or in combination with piracetam (10 µM) for 24 h. Controls were maintained in drug-free medium (without drug exposure). Representative immunostaining images of (A) HDAC1 expression (red) and nuclear staining with DAPI (blue) obtained by confocal microscopy (magnification 100x, scale bar 100 µm). (C) GCN5 expression (red) and nuclear staining with DAPI (blue) obtained by confocal microscopy (magnification 100x). The data are expressed as the mean ± standard error mean of the corrected total cell fluorescence (CTCF) of three independent experiments. Representative graphs depicting immunostaining analysis results showing CTCF values for HDAC1 and GCN5 for all groups in B and D, respectively. N = 3. ****$P < 0.0001$, ***$P < 0.001$, **$P < 0.01$, *$P < 0.05$, NS—nonsignificant.

biogenesis and epigenetic changes [47]. Psychostimulant and opioid abuse induces CNS dysfunctions and ultimately affect behavioral and neurocognitive functions [47]. The differential acetylation of histone proteins is an epigenetic marker of the regulation of chromatin structure and transcriptional status [48]. The interplay between HATs and HDACs maintains the post-translational modifications of the conserved tail amino acids of histone proteins, such as the acetylation of histones [49]. Specific changes in acetylation status are associated with the deregulation of chromatin structure, which has been linked to several human disorders [50]. However, there is still a gap in knowledge regarding cocaine-, METH- and morphine-induced epigenetic modifications of HATs and HDACs (class I and class II HDACs) in the brain. In the present work, we examined the effects of cocaine, METH and morphine on acetylation-regulating HAT and HDAC enzymes and H3KAC as a marker of changes in the global level of acetylation in human primary astrocytes. Moreover, we sought to decipher the potential of the nootropic drug piracetam to exert a neuroprotective effect against the effects of psychostimulant and opioid abuse on essential candidates in the acetylome of human primary astrocytes.

HDAC class I proteins play a major role in deacetylation activity and epigenetic changes associated with psychostimulant abuse [51]. Kumar *et al.* (2005) demonstrated that the acute and chronic exposure of cocaine induces specific histone modifications at gene promoters and modulates HDAC expression, which further alters locomotor and reward responses to cocaine in animal model [52]. Intriguingly, HDAC1 overexpression is linked with neurodegenerative diseases, and the inhibition of HDAC class I proteins with an HDAC inhibitor was indicated to be potentially neuroprotective [53–55]. The first goal of this study was to define the effects of the psychostimulants- cocaine, METH and opioid-morphine on HDAC class I proteins.

Exposure to cocaine and METH significantly upregulated HDAC1 protein expression, and coexposure with piracetam reversed these effects, resulting in expression levels similar to those in the control. We observed a similar trend in HDAC1 mRNA expression levels mediated by the psychostimulants. However, cocaine exposure in astrocytes did not result in any significant changes in HDAC2 and HDAC3 protein levels. This result is quite interesting from the point of view of psychostimulant-induced permanent changes in neural plasticity. Kennedy *et al.* (2013) reported the behavioral changes in animals due to cocaine exposure and they demonstrated that HDAC1 knockdown significantly reduced cocaine locomotor sensitization, whereas the local knockdown of HDAC2 or HDAC3 did not show any effect on cocaine-induced behaviors [32]. Furthermore, immunostaining analysis confirmed that HDAC1 was highly expressed in the nucleus. Cocaine, METH and morphine exposure interfered with the nuclear localization of HDAC1 and induced its translocation to the cytoplasm. Previous studies investigated that under the neurotoxic condition, HDAC1 translocate to cytoplasmic compartment of neurons, whereas the treatment of neurons with calcineurin inhibitors reversed the neurotoxic effect and prevented translocation of HDAC1. Furthermore, it also showed the nuclear localization of HDAC1. This study has mainly shown that nuclear localization of HDAC1 is important for neuroprotection [55]. Additionally, in the present study, coexposure of piracetam with substances of abuse prevented HDAC1 translocation to the cytoplasm [55]. Moreover, METH exposed animals showed decreased levels of HDAC3 and suggested HDAC2 dependent mechanism. The outcomes of this study point towards the changes in HDAC1 levels due to exposure to psychostimulants may result in long-lasting transcriptional changes.

*In vitro* studies have shown that HDAC class II enzymes exhibit substantially lower catalytic activity than HDAC class I enzymes in deacetylation of lysine residues present in histone proteins of nucleosome complex. However, previous animal studies have demonstrated that HDAC class II proteins were significantly altered in nucleus accumbens region of animal brains [56]. The observed results indicate that the exposure of astrocytes to cocaine, METH and morphine resulted in contrasting modification patterns of HDAC class II proteins. HDAC4, HDAC5 and HDAC7 belong to class IIa, while HDAC6 is a class IIb protein. We observed that cocaine and METH exposure in astrocytes upregulated HDAC4, HDAC6 and HDAC7 protein levels, while HDAC5 was downregulated. Moreover, morphine exposure in astrocytes significantly upregulated HDAC4 levels, while other HDAC class II proteins showed downregulation compared to their levels in the control. Interestingly, HDAC4 plays an important role in synaptic plasticity and memory formation. Therefore, the dysregulation of HDAC4 expression has been shown to affect spatial learning as well as synaptic plasticity [57, 58]. We observed a similar trend in the results of HDAC class II gene expression studies when astrocytes were exposed to cocaine, METH and morphine. Interestingly, previous studies have shown that HDAC4 plays important role in locomotory movements and memory function [57–59]. The exposure of cocaine to animals showed that higher level of phosphorylation of HDAC4 which further results in derailing of normal locomotor responses [60]. *In vivo* model studies demonstrated that cocaine exposure decreases HDAC5 function in the NAc; therefore, HDAC5 exerts its repressive action through the acetylation and transcription of HDAC5 target genes. However, HDAC4 shows activity opposite to the repressive action of HDAC5 upon exposure to cocaine, which contribute towards cocaine reward [17]. Moreover, the overexpression of class IIa HDAC5 in the NAc inhibits cocaine-induced conditioned place preference in *in vivo* model study [17].

In the case of HDAC6 gene expression, METH exposure resulted in significant upregulation, and coexposure to METH and piracetam protected HDAC6 mRNA expression, resulting in expression similar to that in the control group. Intriguingly, studies have shown that

administration of METH in endothelial cells regulates cytoplasmic HDAC6 enzyme localization and alters the epigenetic landscape [61]. HDAC6 has been shown to be responsible for the blocking the activity of peroxiredoxin-1 and peroxiredoxin-2 proteins by exerting deacetylation activity on both of these proteins [62, 63]. Moreover, previous rodent model studies aimed at investigating mood disorders and depression have confirmed the elevation of HDAC6 gene expression, whereas HDAC6 inhibitors have been shown to be effective as antidepressants and mood stabilizers [64]. Interestingly, substance abuse disorder patients commonly suffer from psychiatric comorbidities, including depression and bipolar disorders [65]. Therefore, it is imperative to recognize the role of HDAC6 in the effects of psychostimulant abuse at the cellular level, and our study revealed that HDAC6 expression was significantly altered by psychostimulant exposure. Furthermore, this study demonstrated that the HDAC1, HDAC4, HDAC5 and HDAC6 proteins showed remarkable changes in astrocytes following psychostimulant exposure, thus may indicate distinct functions of these enzymes within addiction-associated neurodegenerative cascades.

Earlier studies mainly reported the effect of drug abuse on HDACs level in *in vivo* studies. Previous study has proved that increase in HDAC2 activity in VTA DA neurons and reduced histone H3 acetylation at lysine 9 (Ac-H3K9) in the VTA 24 h after the injection [57]. Moreover, *in vivo* studies have shown that HDAC3 plays negative regulator role in cocaine context -associated memory formation in mice and inhibition of HDAC3 increases extinction of cocaine-seeking behavior [58, 59]. In case of HDAC class II proteins, METH injections decreased HDAC4 and HDAC7 mRNA expression but increased HDAC6 mRNA levels in *in vivo* studies [66]. Moreover, animal studies have shown that cocaine exposure increases HDAC4 expression which further affects in reducing H3K9/14Ac levels and cocaine-induced conditioned place preference [52]. Cocaine exposure subcellular localization of HDAC5 by regulating its phosphorylation, which facilitates nuclear export via phosphorylation [17, 67]. In addition to this, Hdac5 knockout mice exhibited enhanced cocaine- induced conditioned place preference acquisition which underlies the importance of HDAC5 [17].

HATs are essential players in the acetylation of lysine residues that reduce the positive charges of histones and become attracted to the negatively charged backbone of DNA, which makes the chromatin structure more accessible to transcriptional activators, leading to gene transcription [68]. Our investigation of the exposure of human primary astrocytes to psychostimulants revealed that the HATs- PCAF and GCN5 were downregulated, whereas p300 was upregulated. Interestingly, coexposure to piracetam and morphine restored p300, PCAF and GCN5 protein expression to levels similar to those in the control. Earlier studies have mainly focused on the CBP/p300 family of HATs in the brain and have implicated these enzymes in learning, memory and in psychostimulant addiction [69–71]. Studies have shown that cocaine-exposed CBP-deficient mice exhibit increase in histone acetylation and changes in behavioral responses, which indicate the importance of HATs in maintaining chromatin structure [69].

H3 histone protein acetylation is considered to be a marker of activated chromatin and shows strong correlations with the transcription of genes [16]. Earlier studies have shown that cocaine exposure increases global and gene-specific histone H3 acetylation in an animal model [52]. Additionally, genome-wide chromatin immunoprecipitation microarray (ChIP-chip) analyses revealed that the NAc of cocaine-treated mice exhibited increased H3 gene acetylation and showed a significant increase in the global acetylation of H3 proteins [14]. We investigated the effects of cocaine and METH on the global acetylation level of H3 histone sequences with various ε-amino-terminal tails at lysines 9, 14, 18, 27 and 56. We observed an increase in the acetylation levels of H3K9, H3K18, H3K27 and H3K56 but not that of H3K14 compared to the levels in the control. However, morphine exposure upregulated the acetylation of H3K9,

H3K14, H3K18, H3K27, and H3K56, which is consistent with previous studies [21–25]. The global H3 acetylation increase at various lysine residues is consistent with previous findings confirming epigenetic changes due to psychostimulant exposure [52, 72].

The findings of this preliminary study on psychostimulant and opioid-mediated global acetylome changes in human primary astrocytes might contribute to the identification of early candidate HDACs and HATs. This may potentially reveal suitable targets which may help to promote neuroprotection and repair in addiction-associated disorders. Overall, HDACs play a major role in maintaining acetylation levels within cells. However, overexpression of HDACs level may affect acetylation level which may result in chromatin remodeling of nucleosome. Previous studies have shown that HDAC inhibitors have been demonstrated to be beneficial in animal models of neurodegenerative diseases. Some HDAC inhibitors are HDAC class specific, which makes them unsuitable for inhibiting pan-HDACs. However, nootropic agents such as piracetam are well tolerated and were demonstrated to exert protective effects on HDAC1 protein expression in our preliminary studies. Moreover, HDAC5 expression was protected by piracetam from the effects of cocaine and morphine. However, other HDACs and HATs proteins did not show any significant protective effect of piracetam.

## 5. Conclusion

In summary, this *in vitro* study provides a basic understanding of the effect of psychostimulants and opioid exposure on the expression levels of HATs and HDACs in distinctive patterns. We also observed significant changes in acetylation levels at various global H3 histone lysine residues. Additionally, the potential role of the nootropic drug piracetam in protecting the expression levels of HDACs, HATs and the acetylation level of H3K was investigated in human primary astrocytes. We observed that piracetam protected HDAC1 protein expression from the effects of cocaine, METH and morphine. However, to identify which specific HATs and HDACs are responsible for changes in the acetylation levels of histones due to exposure to psychostimulants, further studies in animal models may provide a better picture of acetylome changes.

## Supporting information

**S1 Raw images.**
(PDF)

**S2 Raw images.**
(PDF)

**S1 Dataset.**
(XLSX)

## Author Contributions

**Data curation:** Gurudutt Pendyala, Thangavel Samikkannu.

**Funding acquisition:** Thangavel Samikkannu.

**Investigation:** Mayur Doke, Thangavel Samikkannu.

**Supervision:** Thangavel Samikkannu.

**Validation:** Thangavel Samikkannu.

**Writing – review & editing:** Gurudutt Pendyala, Thangavel Samikkannu.

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
