## [Decision Letter · Decision Letter 0]

11 Feb 2021

PONE-D-21-01012

Psychostimulants and Opioids Differentially Influence the Epigenetic Modification of Histone Acetyltransferase and Histone Deacetylase 

PLOS ONE

Dear Dr. Thangavel

Thank you for submitting your manuscript to PLOS ONE. After careful consideration, we feel that it has merit but does not fully meet PLOS ONE’s publication criteria as it currently stands. Therefore, we invite you to submit a revised version of the manuscript that addresses the points raised during the review process.

ACADEMIC EDITOR:

1. The authors did extensive analyses of  HDACs and HATs in the n vitro set up. However, the authors conclusion was oversimplified. The authors need to perform more experiments to validate the conclusion and precise the discussion and conclusion according to the results. 

2. In vivo studies would strengthen the manuscript. Authors are encouraged to do these and to some extent some mechanisitic studies

We look forward to receiving your revised manuscript.

Kind regards,

Shilpa Buch

Academic Editor

PLOS ONE

Journal Requirements:

2)  Thank you for stating the following financial disclosure:

 [No].

3) Thank you for stating the following in your Competing Interests section: 

[No].

4) In your Data Availability statement, you have not specified where the minimal data set underlying the results described in your manuscript can be found. PLOS defines a study's minimal data set as the underlying data used to reach the conclusions drawn in the manuscript and any additional data required to replicate the reported study findings in their entirety. All PLOS journals require that the minimal data set be made fully available. For more information about our data policy, please see http://journals.plos.org/plosone/s/data-availability.

5) PLOS ONE now requires that authors provide the original uncropped and unadjusted images underlying all blot or gel results reported in a submission’s figures or Supporting Information files. This policy and the journal’s other requirements for blot/gel reporting and figure preparation are described in detail at https://journals.plos.org/plosone/s/figures#loc-blot-and-gel-reporting-requirements and https://journals.plos.org/plosone/s/figures#loc-preparing-figures-from-image-files. When you submit your revised manuscript, please ensure that your figures adhere fully to these guidelines and provide the original underlying images for all blot or gel data reported in your submission. See the following link for instructions on providing the original image data: https://journals.plos.org/plosone/s/figures#loc-original-images-for-blots-and-gels.

6) PLOS requires an ORCID iD for the corresponding author in Editorial Manager on papers submitted after December 6th, 2016. Please ensure that you have an ORCID iD and that it is validated in Editorial Manager. To do this, go to ‘Update my Information’ (in the upper left-hand corner of the main menu), and click on the Fetch/Validate link next to the ORCID field. This will take you to the ORCID site and allow you to create a new iD or authenticate a pre-existing iD in Editorial Manager. Please see the following video for instructions on linking an ORCID iD to your Editorial Manager account: https://www.youtube.com/watch?v=_xcclfuvtxQ

Reviewers' comments:

Reviewer's Responses to Questions

**Comments to the Author**

1. Is the manuscript technically sound, and do the data support the conclusions?

Reviewer #1: Partly

Reviewer #2: Partly

2. Has the statistical analysis been performed appropriately and rigorously? 

Reviewer #1: Yes

Reviewer #2: Yes

3. Have the authors made all data underlying the findings in their manuscript fully available?

Reviewer #1: Yes

Reviewer #2: Yes

4. Is the manuscript presented in an intelligible fashion and written in standard English?

Reviewer #1: Yes

Reviewer #2: Yes

5. Review Comments to the Author

Reviewer #1: The manuscript submitted Mayur Doke et al., entitled “Psychostimulants and Opioids Differentially Influence the Epigenetic Modification of Histone Acetyltransferase and Histone Deacetylase” demonstrated that psychostimulant and opioid exposure significantly modulates the expression of histone acetyl transferases (HDAC) and histone deactylase (HATs) human astrocytes. Further, the authors stated that nootropic drug piracetam prevented psychostimulant effects. The author did extensive analysis HDAC, acetylation, and HATs in in vitro set up. However, the authors conclusion was oversimplified. The authors need to perform more experiments to validate the conclusion and precise the discussion and conclusion according to the results. The following comments are need to be addressed.

Comments

1. The authors analysed well in in vitro, however, It would be more solid conclusion if the author validates these findings in in vivo experimental conditions using rodents’ models.

2. The author also needs to clearly mention about the group comparison in legends, which group compared with which group. Did the author compared cocaine with cocaine+piracetam and similarly for other psychostimulant. If so please mention clearly in legends as well as in results.

3. The authors needs to discuss the roles of other HDACs except HDAC1 in pychostimulant mediated acetylation in astrocytes and its epigenetic modifications. Since most of the HDACs and HAT are not affected by Piracetam treatment.

4. What is the rationale for choosing Piracetam and discuss, how it regulates HDACs in detail.

5. In Figure 3, the results of PCAF, p300 and GCN5 showed no significant difference between psychostimulant with and without piracetam. How the authors concluded piracetam protected psychostimulant mediated effects on astrocytes? This has to be properly mentioned through out the manuscript and has to be discussed.

6. In figure 4, The gene expression analysis of GCN5 was not affected by psychostimulants or piracetam however, the protein expression and ICC in figure 5 showing prominent effects. Please discuss.

7. The authors need to clearly mention in conclusion about the protective effects of piracetam on HDAC and HATs, since most of the modulatory effects of pycostimulants were not prevented by piracetam. However, the author generalized the result conclusion.

8. Did the author expose the cells with all the three psychostimulants together or in two combinations? Does it aggravate the effects on HDAC and HAT as well as its acetylation’s?

9. Please discuss the rationale for choosing the doses of psychostimulants.

10. In Fig 6, the scale bar was missing.

Reviewer #2: In this manuscript, Doke et al. demonstrated the involvement of psychostimulants and opioid-mediated epigenetic changes that occurred at the histone level in human primary astrocytes. The authors also determined the neuroprotective role of piracetam on both the histone deacetylases (HDACs) and histone (HAT) levels in human primary astrocytes exposed to psychostimulants and opioids. Though this study seems very interesting, lack of rigorous experiments and mechanistic approach notably hinders the scientific enthusiasm. Further, most of the experiments were of primitive. Overall, this manuscript needs major revision.

Comments:

1. How does the neuroprotective agent piracetam regulate the expression of HDACs and HATs? Please explain the relevant molecular mechanism.

2. Figure 2A and 2B: missed labeling for the control group.

3. Figure arrangements were not of publication format.

4. More mechanistic studies are warranted in the context of HDACs and HAT inhibitors/activators. Gene silencing approaches would make it more convincing.

5. It would be more compelling if the authors validate these findings in an in vivo model system.

6. Immunostaining data lacks scalebar.

7. It is unclear that piracetam's effects on the combinatorial exposure of all these drugs-cocaine, meth, and morphine.

8. Please provide rationale for choosing the dose concentration of all these drugs.

6. PLOS authors have the option to publish the peer review history of their article (what does this mean?). If published, this will include your full peer review and any attached files.

Reviewer #1: No

Reviewer #2: No

---

## [Author Response · Author response to Decision Letter 0]

15 Mar 2021

Reviewer #1: The manuscript submitted Mayur Doke et al., entitled “Psychostimulants and Opioids Differentially Influence the Epigenetic Modification of Histone Acetyltransferase and Histone Deacetylase” demonstrated that psychostimulant and opioid exposure significantly modulates the expression of histone acetyl transferases (HDAC) and histone deactylase (HATs) human astrocytes. Further, the authors stated that nootropic drug piracetam prevented psychostimulant effects. The author did extensive analysis HDAC, acetylation, and HATs in in vitro set up. However, the authors conclusion was oversimplified. The authors need to perform more experiments to validate the conclusion and precise the discussion and conclusion according to the results. The following comments need to be addressed.

Comments

1. The authors analysed well in in vitro, however, It would be more solid conclusion if the author validates these findings in in vivo experimental conditions using rodents’ models.

Response: We thank the reviewers for this excellent suggestion and do concur validation of our current findings in vivo will further strengthen the manuscript. However, due to the significant restrictions associated with COVID-19, the uncertainty when these will be lifted coupled to the very limited staffing; we unfortunately cannot perform the suggested animal studies. We hope the reviewers can understand this limitation is not from a scientific perspective but merely administrative sanctions at this time. 

2. The author also needs to clearly mention about the group comparison in legends, which group compared with which group. Did the author compared cocaine with cocaine+piracetam and similarly for other psychostimulant. If so please mention clearly in legends as well as in results.

Response- We have performed two-way ANOVA analysis and compared the groups between cocaine with cocaine + piracetam, METH with METH + piracetam and morphine with morphine + piracetam in Western blot and PCR data analysis. We have also mentioned comparison between cocaine with cocaine+piracetam and similarly for other psychostimulant in all figure legends as well as in results section which are highlighted in red color. We have also made changes in statistical analysis section page no. 7, line no. 197-206. 

3. The authors needs to discuss the roles of other HDACs except HDAC1 in psychostimulant mediated acetylation in astrocytes and its epigenetic modifications. Since most of the HDACs and HAT are not affected by Piracetam treatment.

Response - We have discussed the roles of HDACs in psychostimulant mediated acetylation. However, there are not enough evidence available in the context of roles of HDACs in psychostimulant mediated acetylation in astrocytes. We have made necessary changes in discussion part. The necessary changes will be reflected on page no. 17-18, line no. 510-522 and page no. 19, line no. – 555-559. We have made necessary changes in conclusion part page no. 19, line no. – 566-567.

4. What is the rationale for choosing Piracetam and discuss, how it regulates HDACs in detail.

Response- Piracetam is a nootropic drug and used as a cerebral function regulating dietary supplement which, it is claimed, is able to enhance cognition and memory, slow down brain aging, increase blood flow and oxygen to the brain, aid stroke recovery, and improve Alzheimer's, Down syndrome, dementia, and dyslexia, among others [1] Piracetam is a cyclic derivative of GABA. Previous research studies have demonstrated the effectiveness in treating alcoholism or its symptoms [2-7]. Piracetam has been approved in Europe by European Medicines Agency as a supplemental drug to improve memory functions in Alzheimer's patients [1]. Piracetam has also been showed to have positive influence on epileptic patients. Previously, antiepileptic drugs have been tested as histone deacetylase inhibitors. Therefore, we decided to check the effect piracetam on psychostimulant and opioid exposed astrocytes. 

References- 

1. James South, "Piracetam - the original nootropic", International Antiaging Systems

2. Skondia, V. & Kabes, J., "Piracetam in alcoholic psychoses: a double-blind, crossover, placebo controlled study", J Int Med Res 13, (1985) pp.185-187.

3. Buranji I, Skocilic Z, Kozaric-Kovacic D. "Cognitive function in alcoholics in a double-blind study of piracetam, Lijec Vjesn 1990 Mar-Apr;112(3-4):111-4.

4. S Kalmár, Adjuvant therapy with parenteral piracetam in alcohol withdrawal delirium, Orv Hetil (2003) 144: pp.927-30.

5. Dencker SJ, Wilhelmson G, Carlsson E, Bereen FJ. "Piracetam and chlormethiazole in acute alcohol withdrawal: a controlled clinical trial." J Int Med Res 1978;6(5):395-400.

6. Meyer JG, Forst R, Meyer-Wahl L. "Course of alcoholic predelirium during treatment with piracetam: results of serial psychometric tests (author's transl)", Dtsch Med Wochenschr 1979 Jun 22;104(25):911-4

7. Binder S, Doddabela P. "The efficacy of Piracetam on the mental functional capacity of chronic alcoholics (author's transl)", Med Klin 1976 Apr 23;71(17):711-6

8. Marco Fedi, MD; David Reutens, MD, FRACP; François Dubeau, MD, FRCPC; et al. “Long-term Efficacy and Safety of Piracetam in the Treatment of Progressive Myoclonus Epilepsy”.

5. In Figure 3, the results of PCAF, p300 and GCN5 showed no significant difference between psychostimulant with and without piracetam. How the authors concluded piracetam protected psychostimulant mediated effects on astrocytes? This has to be properly mentioned throughout the manuscript and has to be discussed.

Response - We have made necessary changes as per reviewers instruction. PCAF and p300 showed no significant difference between psychostimulant with and without piracetam. However, METH and morphine in morphine with and without piracetam showed slight changes in GCN5 expression. We have removed the generalized statement stating piracetam protected psychostimulant mediated effects on astrocytes and the necessary changes can be seen on page no. 11, line no. – 309-310 and 321-322.

6. In figure 4, The gene expression analysis of GCN5 was not affected by psychostimulants or piracetam however, the protein expression and ICC in figure 5 showing prominent effects. Please discuss.

Response - We understand that gene expression analysis of GCN5 data not completely correlate with the protein expression of GCN5 in western blot and ICC data. But there are some percentage of similarity between gene expression and protein expression of GCN5. The reason could be because of there are many processes between transcription and translation and as mentioned above protein stability is a big factor. The half-life of different proteins can vary a lot. Moreover, gene expression is controlled at transcriptional and post-transcriptional regulation. Protein expression is depending on protein stability and modifications. 

7. The authors need to clearly mention in conclusion about the protective effects of piracetam on HDAC and HATs, since most of the modulatory effects of psychostimulants were not prevented by piracetam. However, the author generalized the result conclusion.

Response - We have made necessary changes in discussion part. The necessary changes will be reflected on page no. 19, line no. – 555-559. We have made necessary changes in conclusion part on page no. 19, line no. – 566-567.

8. Did the author expose the cells with all the three psychostimulants together or in two combinations? Does it aggravate the effects on HDAC and HAT as well as its acetylation’s?

Response - Cocaine and METH are known psychostimulants and morphine is classified under opioids. So, we have not performed any study to check the effect of all three substance of abuse or combination of two on HDAC and HAT as well as its acetylation’s. But, in further study we will definitely study the effect of two psychostimulants. 

9. Please discuss the rationale for choosing the doses of psychostimulants.

Response - Our previous studies have already evaluated range of dosages of psychostimulants cocaine, METH and morphine as well as “nootropic” drug piracetam on human primary astrocytes for toxicity studies [8–11]. Cocaine (1 µM), METH (10 µM), morphine (5 µM) and piracetam (10 µM) did not show any toxicity at particular mentioned concentrations. We have also performed cytotoxicity test with range of concentration of drugs and selected the above-mentioned dosages which is far below than lethal concentration dose of drugs.

References- 

8. Samikkannu T, Ranjith D, Rao KVK, Atluri VSR, Pimentel E, El-Hage N, et al. HIV-1 gp120 and morphine induced oxidative stress: role in cell cycle regulation. Front Microbiol. 2015;6: 614. doi:10.3389/fmicb.2015.00614

9. Sivalingam K, Cirino TJ, McLaughlin JP, Samikkannu T. HIV-Tat and Cocaine Impact Brain Energy Metabolism: Redox Modification and Mitochondrial Biogenesis Influence NRF Transcription-Mediated Neurodegeneration. Mol Neurobiol. 2020; 1–15. doi:10.1007/s12035-020-02131-w

10. Sivalingam K, Samikkannu T. Neuroprotective Effect of Piracetam against Cocaine-Induced Neuro Epigenetic Modification of DNA Methylation in Astrocytes. Brain Sci. 2020;10: 611. doi:10.3390/brainsci10090611

11. Doke M, Jeganathan V, McLaughlin JP, Samikkannu T. HIV-1 Tat and cocaine impact mitochondrial epigenetics: effects on DNA methylation. Epigenetics. 2020 [cited 17 Dec 2020]. doi:10.1080/15592294.2020.1834919

10. In Fig 6, the scale bar was missing.

Response - Thank you for notifying our mistake. We have added scale bar in figure legend as well as some of the representative immunofluorescence images. 

Reviewer #2: In this manuscript, Doke et al. demonstrated the involvement of psychostimulants and opioid-mediated epigenetic changes that occurred at the histone level in human primary astrocytes. The authors also determined the neuroprotective role of piracetam on both the histone deacetylases (HDACs) and histone (HAT) levels in human primary astrocytes exposed to psychostimulants and opioids. Though this study seems very interesting, lack of rigorous experiments and mechanistic approach notably hinders the scientific enthusiasm. Further, most of the experiments were of primitive. Overall, this manuscript needs major revision.

Comments:

1. How does the neuroprotective agent piracetam regulate the expression of HDACs and HATs? Please explain the relevant molecular mechanism.

Response - The mode of action of piracetam in human body is not completely know. However, some research studies had shed some light on the mechanism action of piracetam and hypothesized that it acts on ion channels or ion carriers, thus leading to non-specific increased neuron excitability. Moreover, it has been shown to increase blood flow and oxygen consumption in parts of the brain [1,2]. Neurotransmitter acetylcholine can be improved with piracetam by using the muscarinic cholinergic (ACh) receptors, which directly influence memory processes [3]. Piracetam may also have an effect on NMDA glutamate receptors, which is also involved in learning and memory processes. Piracetam is theorized to increase cell membrane permeability [3,4] and exert a global effect on brain neurotransmission by modulation of ion channels such as Na+ and K+ [1]. It has been proven that increase of oxygen consumption to the brain has a direct affect in ATP metabolism and will increase activity of adenylate kinase in a rat brain [5,6]. Piracetam is thought to increase synthesis of cytochrome b5 [7], thus playing a role in electron transport mechanism in mitochondria. It has also been proven to increase permeability in mitochondria to some intermediates of the Krebs cycle [8].

References-

1. AH Gouliaev, A Senning, "Piracetam and other structurally related nootropics" Brain Research. Brain Research Reviews. 1994 May;19(2):180-222.

2. "Cerebral blood flow effects of piracetam, pentifylline, and nicotinic acid in the baboon model compared with the known effect of acetazolamide." Arzneimittelforschung. 1996 Sep;46(9):844-7.

3. "Piracetam--an old drug with novel properties?" Acta Pol Pharm. 2005 Sep-Oct;62(5):405-9.

4. "Piracetam: novelty in a unique mode of action." Pharmacopsychiatry. 1999 Mar;32 Suppl 1:2-9.

5. Grau, M. et al, "Effect of Piracetam on electrocorticogram and local cerebral glucose utilization in the rat", Gen Pharmac 18 (1987) pp.205-211.

6. Nickolson, V. & Wolthuis, 0., "Effect of the acquisition - enhancing drug Piracetam on rat cerebral energy metabolism", Biochem Pharmacol 25, (1976) pp.2241-2244.

7. Tacconi, M. & Wurtman, R. "Piracetam: physiological disposition and mechanism of action" in Advances in Neurology, vol. 43 (1986) S. Fahn et al, ed. Raven Press: NY.

8. Grau, M. et al, "Effect of Piracetam on electrocorticogram and local cerebral glucose utilization in the rat", Gen Pharmac 18 (1987), pp. 205-211

2. Figure 2A and 2B: missed labeling for the control group.

Response - We have added missing labels for figures 2A and 2B. 

3. Figure arrangements were not of publication format.

Response - As per your suggestion, we have modified our figures and they are now in publication format.

4. More mechanistic studies are warranted in the context of HDACs and HAT inhibitors/activators. Gene silencing approaches would make it more convincing.

Response - Thank you for your suggestions. In this study, primarily we wanted to study the effects of psychostimulants and opioids on acetylation regulating proteins. Therefore, we studied effects of psychostimulants and opioids on HDACs and HAT. In future studies, we will plan to include more mechanistic studies consisting of the gene silencing approach.

5. It would be more compelling if the authors validate these findings in an in vivo model system.

Response: We thank the reviewers for this excellent suggestion and do concur validation of our current findings in vivo will further strengthen the manuscript. However, due to the significant restrictions associated with COVID-19, the uncertainty when these will be lifted coupled to the very limited staffing, we unfortunately cannot perform the suggested animal studies. We hope the reviewers can understand this limitation is not from a scientific perspective but merely administrative sanctions at this time. 

6. Immunostaining data lacks scale bar.

Response - Thank you for notifying our mistake. We have added scale bar in figure legend as well as some of the representative immunofluorescence images.

7. It is unclear that piracetam's effects on the combinatorial exposure of all these drugs-cocaine, meth, and morphine.

Response - In order to understand the combinatorial exposure of all these drugs-cocaine, meth, and morphine on HDACs and HAT, we performed two-way ANOVA analysis and demonstrated the effect of combinatorial exposure on HDACs and HAT proteins. We have compared combinatorial exposure of all these drugs-cocaine, meth, and morphine with piracetam against drugs-cocaine, meth, and morphine alone which we depicted in bar graphs.

8. Please provide rationale for choosing the dose concentration of all these drugs.

 Response - Our previous studies have already evaluated range of dosages of psychostimulants cocaine, METH and morphine as well as “nootropic” drug piracetam on human primary astrocytes for toxicity studies [9–12]. Cocaine (1 µM), METH (10 µM), morphine (5 µM) and piracetam (10 µM) did not show any toxicity at particular mentioned concentrations. We have also performed cytotoxicity test with range of concentration of drugs and selected the above-mentioned dosages which is far below than lethal concentration dose of drugs. 

References-

9. Samikkannu T, Ranjith D, Rao KVK, Atluri VSR, Pimentel E, El-Hage N, et al. HIV-1 gp120 and morphine induced oxidative stress: role in cell cycle regulation. Front Microbiol. 2015;6: 614. doi:10.3389/fmicb.2015.00614

10. Sivalingam K, Cirino TJ, McLaughlin JP, Samikkannu T. HIV-Tat and Cocaine Impact Brain Energy Metabolism: Redox Modification and Mitochondrial Biogenesis Influence NRF Transcription-Mediated Neurodegeneration. Mol Neurobiol. 2020; 1–15. doi:10.1007/s12035-020-02131-w

11. Sivalingam K, Samikkannu T. Neuroprotective Effect of Piracetam against Cocaine-Induced Neuro Epigenetic Modification of DNA Methylation in Astrocytes. Brain Sci. 2020;10: 611. doi:10.3390/brainsci10090611

12. Doke M, Jeganathan V, McLaughlin JP, Samikkannu T. HIV-1 Tat and cocaine impact mitochondrial epigenetics: effects on DNA methylation. Epigenetics. 2020 [cited 17 Dec 2020]. doi:10.1080/15592294.2020.1834919

---

## [Decision Letter · Decision Letter 1]

25 May 2021

Psychostimulants and Opioids Differentially Influence the Epigenetic Modification of Histone Acetyltransferase and Histone Deacetylase

PONE-D-21-01012R1

Dear Dr. Thangavel,

We’re pleased to inform you that your manuscript has been judged scientifically suitable for publication and will be formally accepted for publication once it meets all outstanding technical requirements.

Kind regards,

Axel Imhof

Academic Editor

PLOS ONE

Additional Editor Comments (optional):

Reviewers' comments:

Reviewer's Responses to Questions

**Comments to the Author**

1. If the authors have adequately addressed your comments raised in a previous round of review and you feel that this manuscript is now acceptable for publication, you may indicate that here to bypass the “Comments to the Author” section, enter your conflict of interest statement in the “Confidential to Editor” section, and submit your "Accept" recommendation.

Reviewer #1: All comments have been addressed

Reviewer #2: All comments have been addressed

2. Is the manuscript technically sound, and do the data support the conclusions?

Reviewer #1: Yes

Reviewer #2: Yes

3. Has the statistical analysis been performed appropriately and rigorously? 

Reviewer #1: Yes

Reviewer #2: Yes

4. Have the authors made all data underlying the findings in their manuscript fully available?

Reviewer #1: Yes

Reviewer #2: Yes

5. Is the manuscript presented in an intelligible fashion and written in standard English?

Reviewer #1: Yes

Reviewer #2: Yes

6. Review Comments to the Author

Reviewer #1: The authors addressed all the comments raised by the reviewer point by point. I recommend to accept and publish the article in the present form.

thank you

Reviewer #2: The authors revised the manuscript satisfactorily. Also they mentioned that due to covid-19 restrictions, they couldn't perform some experiments.

7. PLOS authors have the option to publish the peer review history of their article (what does this mean?). If published, this will include your full peer review and any attached files.

Reviewer #1: No

Reviewer #2: No

---

## [Editor Report · Acceptance letter]

3 Jun 2021

PONE-D-21-01012R1 

Psychostimulants and Opioids Differentially Influence the Epigenetic Modification of Histone Acetyltransferase and Histone Deacetylase in astrocytes 

Dear Dr. Samikkannu:

I'm pleased to inform you that your manuscript has been deemed suitable for publication in PLOS ONE. Congratulations! Your manuscript is now with our production department. 

Kind regards, 

on behalf of

Dr. Axel Imhof 

Academic Editor

PLOS ONE